# Rich complex behaviour of self-assembled nanoparticles far from equilibrium

Serim Ilday[1], Ghaith Makey[1], Gursoy B. Akguc[1], Özgün Yavuz[2], Onur Tokel[1], Ihor Pavlov[1], Oguz Gülseren[1] & F. Ömer Ilday[1,2]

A profoundly fundamental question at the interface between physics and biology remains open: what are the minimum requirements for emergence of complex behaviour from non-living systems? Here, we address this question and report complex behaviour of tens to thousands of colloidal nanoparticles in a system designed to be as plain as possible: the system is driven far from equilibrium by ultrafast laser pulses that create spatiotemporal temperature gradients, inducing Marangoni flow that drags particles towards aggregation; strong Brownian motion, used as source of fluctuations, opposes aggregation. Nonlinear feedback mechanisms naturally arise between flow, aggregate and Brownian motion, allowing fast external control with minimal intervention. Consequently, complex behaviour, analogous to those seen in living organisms, emerges, whereby aggregates can self-sustain, self-regulate, self-replicate, self-heal and can be transferred from one location to another, all within seconds. Aggregates can comprise only one pattern or bifurcated patterns can coexist, compete, endure or perish.

[1] Department of Physics, Bilkent University, Ankara 06800, Turkey. [2] Department of Electrical and Electronics Engineering, Bilkent University, Ankara 06800, Turkey. Correspondence and requests for materials should be addressed to S.I. (email: serim@bilkent.edu.tr).

Order, diversity and functionality spontaneously emerge in nature, resulting in hierarchical organization in far-from-equilibrium conditions through stochastic processes, typically regulated by nonlinear feedback mechanisms[1,2]. However, current understanding of the fundamental mechanisms and availability of experimental tools to test emerging theories on the subject are lacking. Most current understanding is from model systems[3–5] that are either too simple to generate rich, complex dynamics collectively[2] or so artificial that they have little relevance to actual physical systems. On the other hand, real-life systems, living organisms being the ultimate examples, are so complicated that it is difficult to isolate the essential factors for emergence of complex dynamics[1,2]. Specific instances of characteristically life-like properties, such as self-replication or self-healing, have been demonstrated in various microscopic systems[6–11], but they were never observed collectively in a single system that is simple enough to allow identification of mechanisms of emergence.

Dissipative self-assembly is a practical experimental platform to study the fundamental mechanisms of emergent complex behaviour by providing settings akin to those found in nature: far-from-equilibrium conditions[12–16], a time-varying external energy input[12–17], nonlinear feedback mechanisms[16,18–22], fast kinetics[15,16,22,23], spatiotemporal control[15,16,22,23] and a medium to efficiently dissipate the absorbed energy[12–17]. However, previous experimental demonstrations either relied on specific interactions between the building blocks and the external energy source[24–26] or were limited to certain materials and/or sizes[21,26–28]. Furthermore, most of them were strongly limited by their slow kinetics[14,29] and there was little room for fluctuations (Brownian motion was usually weak), where the nonlinear feedback mechanisms were often neglected, unemployed or unidentified.

Here, we report far-from-equilibrium self-assembly of tens to thousands of colloidal nanoparticles with fast kinetics that exhibits complex behaviour, analogous to those commonly associated with living organisms, namely, autocatalysis and self-regulation, competition and self-replication, adaptation and self-healing and motility. We do not use functionalized particles or commonly employed interaction mechanisms, such as optical trapping, tweezing, chemical or magnetic interactions. Instead, we designed a simple system that brings together the essential features: nonlinearity to give rise to multiple fixed points in phase space (hence, possibility of multiple steady states), each corresponding to a different pattern and their bifurcations[2]; positive and negative feedback to cause exponential growth of perturbations and their suppression, respectively[18,19,22]; fluctuations to spontaneously induce transitions through bifurcations[1]; and finally, spatiotemporal gradients to drive the system far from equilibrium, whereby the spatial part allows regions with different fixed points to coexist and the temporal part leads to dynamic growth or shrinkage of these regions.

## Results

**Formation of the aggregates**. The experimental system is illustrated in Fig. 1a, where a quasi-two-dimensional (2D; thickness of 1–2 μm) colloidal solution of strongly Brownian polystyrene nanospheres (500 nm in diameter) is sandwiched between two thin microscope slides. Ultrafast laser pulses are focused to a spot size of ~10 μm within the solution (Supplementary Fig. 1). All of these materials are optically transparent at the laser wavelength of 1 μm, and hence energy intake is based on multi-photon absorption of the femtosecond pulses[30] that induces steep spatiotemporal thermal gradients. Localized heat deposition creates an air bubble[31–33] and sets up Marangoni flow[12,34].

This flow drags the particles towards the bubble, which serves as a physical boundary, inducing aggregation.

We first focus on the fluid dynamics and numerically analyse the flow patterns (Fig. 1b and Supplementary Fig. 2). The velocities and trajectories of strongly Brownian particles with respect to this flow are simulated (Supplementary Figs 2 and 3). As expected, the velocity is lowest in the small ellipsoidal area surrounding the bubble, carrying large numbers of particles towards this low-velocity region. In this high-density region, interparticle collision rate increases, overcoming Brownian motion and resulting in aggregation at the bubble boundary (Fig. 1c for numerical simulations). As the aggregate grows, this low-velocity region extends outward and the flow speeds up (Fig. 1d and Supplementary Movie 1). This constitutes a positive feedback mechanism, analogous to auto-catalysis processes[7,35,36] associated with chemical systems, whereby the aggregate can self-sustain (Fig. 1e).

**Toy model of the feedback mechanism**. We developed a toy model (see Methods) to help qualitatively understand the feedback mechanisms that create the self-sustaining aggregate: we focus on a finite area, where an initial aggregation is already forming and introduce the filling ratio, $\phi$, as an order parameter ($\phi = 0$, empty and $\phi = 1$, maximum packing). The fluid flux is similarly described by $\theta$. Assuming laminar flow and permeability to be proportional to $1/\phi^3$, $\phi(t)$ and $\theta(t)$ constitute a 2D dynamic system. If we also assume that the fluid responds to changes in aggregation much faster than vice versa, this 2D system reduces to $\dot{\phi} = \left( \phi - F\phi^{-3} - \langle \xi(t) \rangle_{\mathrm{rms}} \right)\left( 1 - \phi - \langle \xi(t) \rangle_{\mathrm{rms}} \right)$. Here, $F$ and $\langle \xi(t) \rangle_{\mathrm{rms}}$ are normalized flow rate and averaged Brownian motion, respectively. Typical behaviour of this system is depicted for the 2D system in Supplementary Fig. 4 and for the one-dimensional (1D) version in Fig. 1f. Linearized stability analysis[2] shows that the system supports a stable (attracting) fixed point at high $\phi$, corresponding to aggregation and an unstable fixed point at low $\phi$ that serves as a critical point: if the initial value of $\phi$ exceeds this critical value, then the aggregate grows, reducing the fluid flux, $\theta$, that promotes further growth. Otherwise, the drag effect and Brownian motion prevent aggregation. This result explains why aggregates do not form spontaneously, but require a seed that we provide experimentally by creating a bubble. The temporal evolution of $\phi(t)$ matches a sigmoid function (Fig. 1g) that is commonly associated with autocatalytic reactions[7,35,36]. This positive feedback is accompanied by a simultaneously occurring competing feedback mechanism, formally analogous to reaction–diffusion systems[13,14,35,36] (see Methods), between the fluid flow and Brownian motion: the former helps form and reinforce the aggregate, and the latter is dispersive in nature, regulating its growth (Fig. 1e).

**Fast assembly–disassembly experiments**. The scenario described by the toy model is experimentally verified by time-lapse images extracted from Supplementary Movie 2 as shown in Fig. 2a: upon turning the laser on ($t = 0$ s), a bubble forms immediately along with a Marangoni flow ($t = 1$ s) that drags the particles towards the bubble boundary, where they accumulate and form a large aggregate within seconds ($t = 15$ s). Due to this drag force, a region that is fully depleted of particles forms around the bubble. We then turn the laser off at $t = 45$ s and wait for the aggregate to disintegrate ($t = 55$ s), then turn it on again and the aggregate self-assembles largely from the same group of particles, at the same location ($t = 70$ s). For smaller number of particles within the aggregate, much faster ($<1$ s) form–break–reform can also observed in Fig. 2b (time-lapse images from Supplementary Movie 3) when the laser, denoted by the red dot, is turned on and off. This sequence of form–break–reform can be repeated indefinitely, as can be

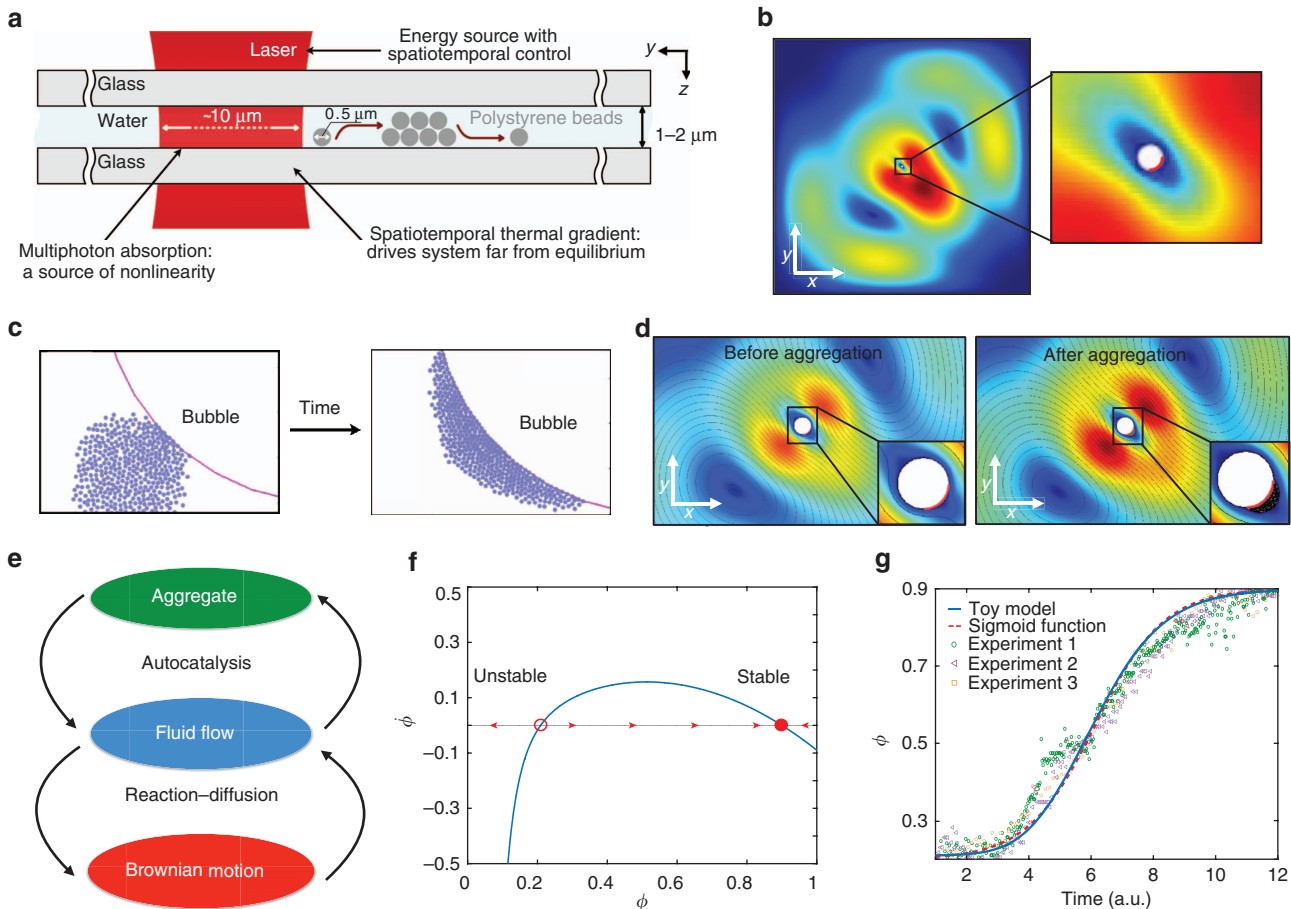

**Figure 1 | Experimental setup and the toy model.** (**a**) Illustration showing colloidal solution of polystyrene spheres sandwiched between two thin microscope glass slides with an ultrafast laser beam focused to ∼10 μm. (**b**) Image displaying velocity field simulation of Marangoni-type microfluidic flow, where red and dark blue areas denote highest and lowest flow speeds, respectively. Simulated area is a 1 cm by 1 cm cell and a bubble of 50 μm diameter is located at the centre of this cell. Magnified image shows that the laser is introduced as a boundary heat source at the lower right quarter of the bubble, depicted by a red line. (**c**) Image showing numerical simulation of the Brownian nanoparticles that are released from a location close to the bubble and aggregate at its boundary. (**d**) Images showing velocity field simulations of the flow before and after an aggregate forms, where the black lines are streamlines. The dark area on the right, magnified image denotes the self-assembled aggregate. (**e**) Schematic description of the nonlinear feedback mechanisms. (**f**) Plot of $\dot{\phi}$ as a function of $\phi$ (filling ratio), showing stable and unstable fixed points for $F = 0.001$ and $\langle \xi(t) \rangle_{rms} = 0.1$. (**g**) Plot comparing toy model and three measurements with the sigmoid function, confirming the autocatalysis characteristics. Experimental data are extracted from the temporal evolution of number of particles in a selected region while forming an aggregate. Toy model data are the evolution of $\phi$ over time (blue line) for $F = 0.001$ and $\langle \xi(t) \rangle_{rms} = 0.1$ with the initial condition of $\phi(0) = 0.21$, fitted with a sigmoid function (red line) of the general form $1/(1 + e^{-t})$.

observed for a number of times in Supplementary Movies 2 and 3. By controllably changing the laser power in the experiments, we can obtain giant aggregates comprising thousands of particles (Fig. 2c) or small clusters (Fig. 2d). Coloured images show calculated Lindemann parameter[37,38], where 0 (blue) means that the neighbouring beads are at their close-packing arrangement, representing solid phase, whereas 1 (red) means that they are distant and independent of each other, corresponding to gas phase (see Supplementary Method 3b).

**Self-regulation of the aggregates.** Moreover, these aggregates can self-regulate in a dynamic environment as shown in Supplementary Movies 4 and 5: Supplementary Movie 4 shows that the aggregates in a diluted (left frame) and in a dense (right frame) colloidal solution are self-regulating to maintain their overall size in a dynamical environment. Left frame shows that the flow constantly carries new particles towards the aggregate. These particles are expected to join in and further enlarge the

aggregate, yet this does not happen since strong Brownian motion of the particles (negative feedback) regulates this tendency and the overall aggregate size is maintained. Similarly, the right frame shows no increase in aggregate size even in a highly dense solution, where jamming of the particles are expected to cause further growth of the aggregate. However, negative feedback again regulates this effect and helps maintain the overall aggregate size. Supplementary Movie 5 shows self-regulation in a more visibly dynamic environment: the movie starts with an already formed aggregate at the boundary of a small bubble ($t = 0$ s). By increasing the laser power, we initiate the growth of the bubble and the aggregate size ($t = 15$ s). Then, by moving the laser beam, we enlarge the bubble but the average size of the aggregate is maintained during this period ($t = 82$ s). Even if we further accelerate the fluid flow, self-regulation mechanism is active and prevents further growth of the aggregate ($t = 142$ s). We also deliberately change the focus of the objective to verify that the aggregate size does not change from one layer to another ($105\,s < t < 130\,s$). Finally, by repositioning the laser beam and

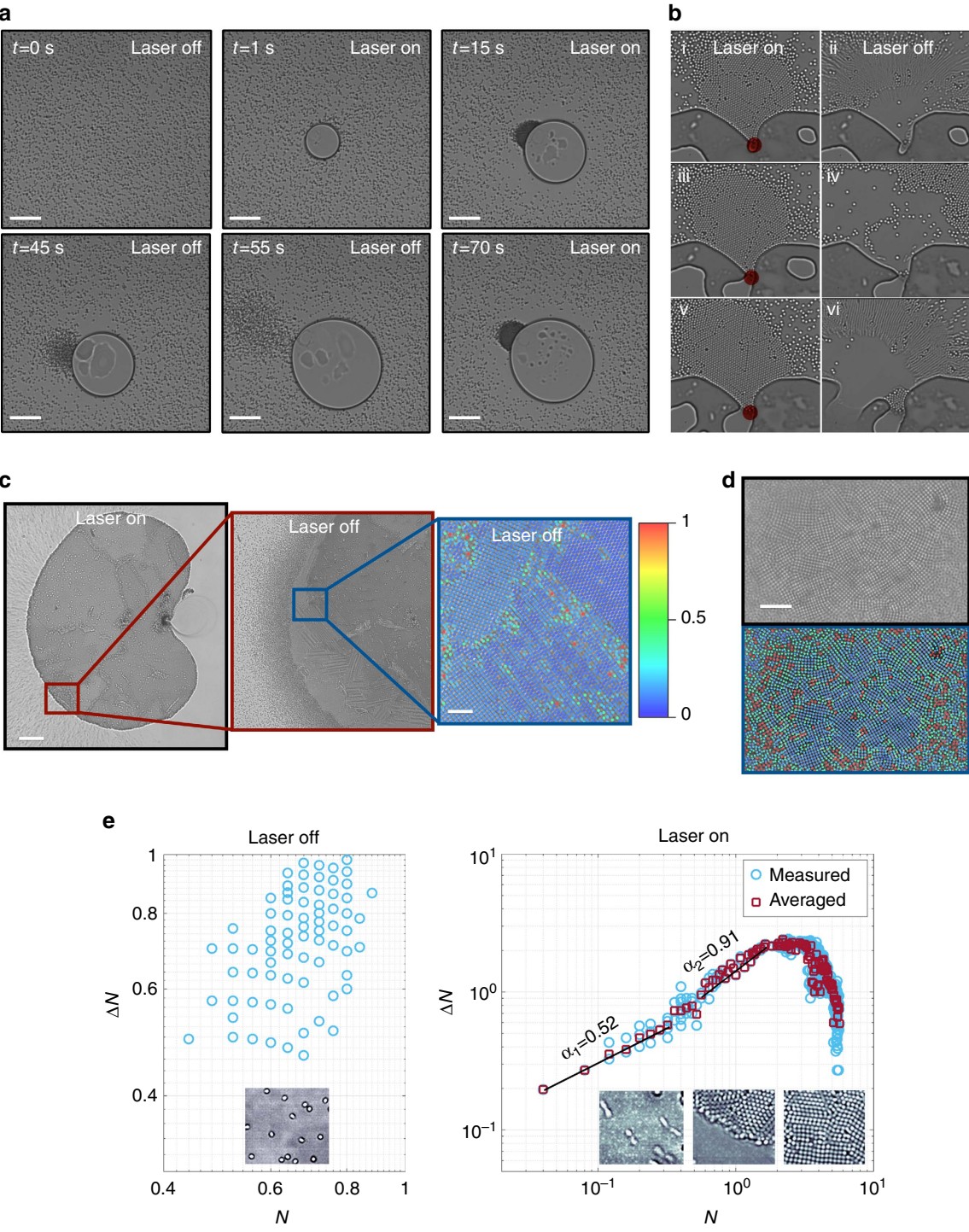

**Figure 2 | Form–break–reform at far-from-equilibrium conditions.** Time-lapse images showing (**a**) that an aggregate can form–break–reform upon turning on and off the laser. Length of the scale bars are 40 μm. (**b**) Form–break–reform behaviour of an aggregate in <1 s, where the red dots denote the laser beam. Images showing (**c**) a large colloidal crystal of square lattice comprising thousands of particles and (**d**) a small cluster of a square lattice with many grains. Coloured images are processed via the Lindemann parameter. (**e**) Plots demonstrating giant number fluctuations analyses under 'laser off' and 'laser on' conditions, where ΔN is the fluctuations and N is the number of particles in a selected region. Lengths of the scale bars are 40 μm for (**a**), 100 μm for the left and 4 μm for the right frame of (**c**) and 5 μm for (**d**).

decreasing the laser power, we shrink the bubble and show that self-regulation still holds ($t = 143$ s).

**Far-from-equilibrium analysis**. To verify that the laser drives this system far from equilibrium, we checked for the presence of giant number fluctuations[25,39] under 'laser off' and 'laser on' conditions (see Fig. 2e and Supplementary Method 3c). As expected, when the laser is off, the central limit theorem applies and normal fluctuations are observed, because the particles are at or near thermal equilibrium, undergoing random (Brownian) motion, where $\Delta N$, the fluctuations, is independent of $N$, the

number of particles in a selected region. Upon turning on the laser, the particles are accelerated and dragged by the flow, where the slope of $N$ initially increases to $\alpha_1 = 0.52$ and then to $\alpha_2 = 0.91$, by the time the aggregate covers half of the selected area, clearly exhibiting giant number fluctuations and confirming that the system is far from equilibrium. Upon filling the selected area by an aggregate, slope decreases sharply as expected. Similarly, the temporal evolution of $N$ has a sigmoid shape, experimentally confirming the autocatalysis dynamics (Fig. 1g)[7,35,36].

**Aggregates with multi- and mono-stable patterns.** Next, we show that the aggregates form colloidal crystals with different symmetries, namely, square (Fig. 3a), hexagonal (Fig. 3b,c2 and d2), oblique (Fig. 3c1,d1) lattices and Moiré patterns (see Fig. 3d3 and Supplementary Method 3d), that are identified using Lindemann parameter, pair correlation function and reciprocal lattice analyses (Supplementary Figs 6,9 and 10). These crystals can be monostable, comprising only one pattern (Fig. 3a,b) or multistable, where bifurcated patterns can coexist (Fig. 3c,d). When the conditions change (for example, laser power/position, the shape/size of the bubble boundary, and hence the flow and the local particle density), these lattices dynamically change into one another as can be seen from Supplementary Movie 6 and from the time-lapse images in Fig. 4a. Supplementary Movie 6 begins with a monostable crystal of hexagonal lattice forming between two bubbles ($t = 70\,\text{s}$) that transforms into a multistable crystal of hexagonal lattice and Moiré pattern upon shrinkage of the bubble on the right ($t = 90\,\text{s}$). When this bubble is fully deflated ($t = 150\,\text{s}$), Moiré pattern becomes the favoured of the two competing patterns that gradually converts the hexagonal lattice into itself ($t = 170\,\text{s}$) and fills up the available area ($t = 200\,\text{s}$)

(Fig. 4a). In other words, the hexagonal lattice dies and the Moiré pattern survives the competition and self-replicates.

**Self-replication of the aggregates.** Here, self-replication refers to a structure making identical copies of itself on an adjacent region[6–8] as described for cellular automata by von Neumann[3]. If the system was in thermodynamic equilibrium or near equilibrium, we would not have regarded it as self-replication but as crystal growth. In our case, growth of the aggregate is one of many possible, qualitatively distinct outcomes that include conversion to multiple other patterns: in a dynamic system with many kinetic traps, different patterns can coexist and compete, where propagation of the replication information for one of the species must lead to the degradation of the rest of the competing species (by destabilizing their kinetic traps) and amplification of the remaining species (by promoting one stable kinetic trap)[8,40–45]. Moreover, we also show self-replication of a 'daughter' aggregate from the 'mother' aggregate in Supplementary Movie 7: the left frame shows that a bubble forms and Marangoni flow drags the particles towards its boundary to form an aggregate. Then, a second bubble forms and separates the aggregate into two aggregates with the same pattern. Similarly, the right frame shows that part of an already formed large aggregate is being detached and carried to the boundary of another bubble to form the same pattern. Furthermore, our observation is not limited to the hexagonal lattice and the Moiré pattern; we observed self-replication of square lattice and its competition with the hexagonal lattice (Fig. 4b).

**Self-healing of the aggregates.** Moreover, these dynamic patterns demonstrate adaptation or self-healing in response to their changing environment, depending on how strongly perturbed

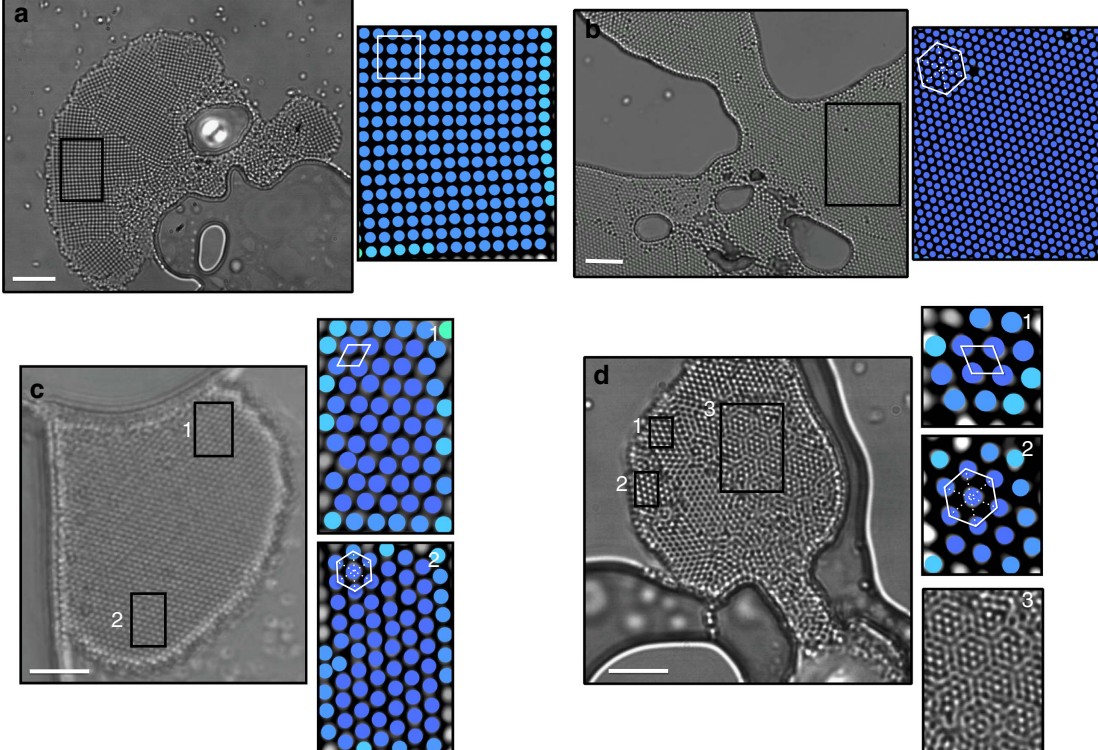

**Figure 3 | Mono- and multi-stable (bifurcated) pattern formation.** Microscope images showing colloidal crystals of monostable (**a**) square and (**b**) hexagonal lattices and multistable (**c**) oblique (1) and hexagonal (2) lattices and (**d**) oblique (1) and hexagonal (2) lattices, and Moiré patterns (3). Colour images are processed images using Lindemann analyses. Geometric shapes are hand drawn to describe the lattice type. Lengths of the scale bars are 5 μm.

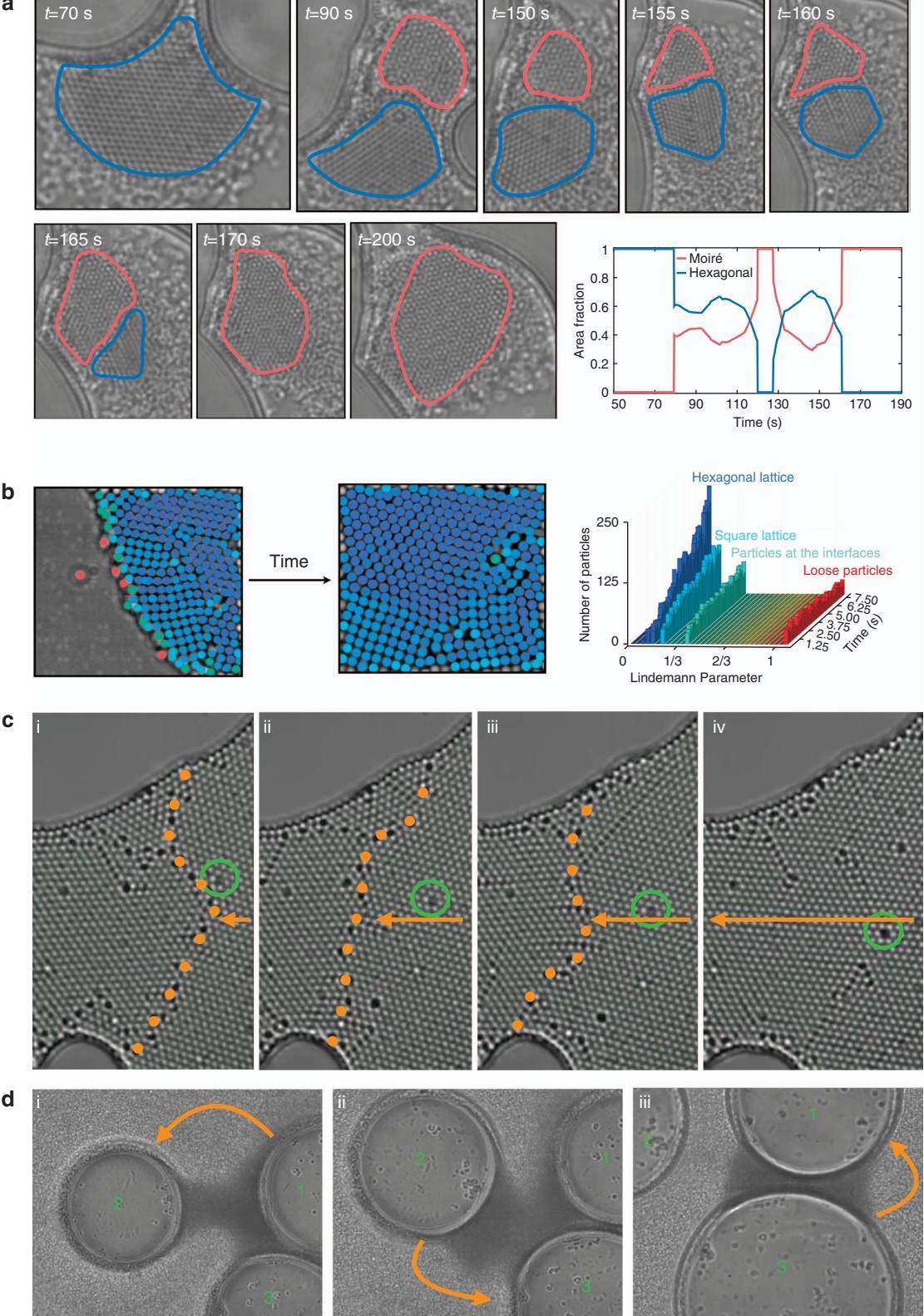

**Figure 4 | Life-like properties. (a)** Time-lapse images showing competition between a hexagonal lattice (within blue boundary) and a Moiré pattern (within red boundary), where the Moiré pattern eventually self-replicates in expense of the hexagonal lattice. Plot showing time evolution of the normalized areas of the two competing patterns. **(b)** Images showing competition between hexagonal (blue) and square (cyan) lattices in a selected area, where the hexagonal lattice self-replicates faster than the square lattice and fills a large portion of the selected area. Colour coding denotes the Lindemann parameter of both lattice types. Plot showing evolution of the Lindemann parameter corresponding to the competing lattices. **(c)** Time-lapse images showing self-healing of a hexagonal lattice under weak perturbations, where the lattice repeatedly anneals out the lattice imperfections (orange dotted line shows a line defect and green circle shows a point defect) to maintain its original pattern. **(d)** Time-lapse images showing motility of the aggregate, where, by repositioning the laser beam, the fluid flow can be altered, resulting in controllable transport of the aggregates from one bubble to another.

they are[12,46,47]: the perturbations insert 'errors' into an ordered structure. If the perturbations are strong enough, the pattern will 'adapt' by transitioning to a different steady state. If the perturbations are weak, the pattern will correct the errors and 'self-heal'. The latter is qualitatively similar to defect annihilation mechanism observed in crystals and error-correction mechanisms in biological organisms. We have already demonstrated how the pattern 'adapts' to new conditions under strong perturbations in Supplementary Movie 6 and in Fig. 4a. Similarly, Supplementary Movie 8 and Fig. 4c show self-healing under weak perturbations, where the hexagonal lattice repeatedly anneals out the lattice imperfections (orange dotted line shows a line defect and green circle shows a point defect) to maintain its original pattern. In Supplementary Movie 9, we also show self-healing ability in a square lattice that has been heavily damaged: the movie starts with a large square lattice. To introduce lattice imperfections, we start to move the laser beam inside the bubble. This results in distorting the shape of the bubble and enlarging it ($t = 33$ s). Then, we move the laser beam again and introduce a second bubble to squeeze the pattern in a wedge ($t = 34$ s) that totally disrupts the square lattice ($t = 49$ s). Finally, we release the squeezed pattern by detaching the second bubble from the first one (by moving the laser beam) and the square pattern self-healed the strain-induced defects and fully recovered.

**Motility of the aggregates.** Finally, we show that by simply repositioning the laser beam and adjusting its power, we create bubbles of predetermined size, shape and position (see Supplementary Movie 10 and Supplementary Methods) and alter the flow, allowing us to transport the aggregates from one bubble to the other. In other words, the aggregates can be rendered motile (Supplementary Movie 11 and Fig. 4d).

## Discussion

Emergence of complex behaviour from this plain system can be understood intuitively under the guidance of our toy model, numerical simulations and experimental observations. The laser-sustained thermal gradient not only keeps the system away from thermal equilibrium but, together with the boundary conditions imposed by the bubbles, also creates different local conditions corresponding to different fixed points: a given location can support, say, a square lattice of self-assembled particles, while a hexagonal lattice exists nearby. Each fixed point has a finite basin of attraction both in the phase space and real space, delineated by the spatially varying conditions. In response to perturbations, such as a shift of a bubble boundary or the omnipresent Brownian motion, the original pattern is recovered (self-healing) if the disturbed state remains within the basin of attraction. If the perturbation is large enough that the disturbed state falls outside of the basin of attraction, it switches to a different pattern (self-adaptation) or can be disassembled. A spatiotemporal gradient can also enlarge or shrink the region where a given pattern is the fixed point. In the former case, the pattern can grow (self-replication) or sustain itself (self-regulation). When two nearby regions supporting different patterns come into contact, competition ensues at their boundary: Brownian motion acting on each particle can displace it just enough that the particle leaves a pattern and joins the adjacent one if this stochastic perturbation is large enough and in the right direction. Consequently, the pattern boundaries are dynamic and if the conditions are favourable, one pattern can grow at the expense of another, demonstrating an analogue of interspecies competition. Similarly, motility can be understood as arising in response to temporal gradients that are small enough that the self-healing property can hold the aggregate together as it moves.

In conclusion, we provide a simple experimental platform for far-from-equilibrium self-assembly to investigate and control a rich set of complex phenomena. We demonstrated collective control of large groups of nanoparticles using only two parameters, the laser power and the beam position, that act at much larger spatial ($\sim 10 \, \mu$m) and temporal (few seconds) scales than those of the individual particles (0.5 $\mu$m, milliseconds) being controlled. Although we worked with 500 nm particles allowing for real-time optical imaging, the scaling of the thermal gradients and strength of Brownian motion with beam and particle size, respectively, suggest the possibility of controlled self-assembly of sub-10 nm nanoparticles using a diffraction-limited laser beam ($\sim 250$ nm). In principle, the methodology can also be applied to other types of materials, nonliving and living alike, with different shapes and properties. The possibility of studying such materials under far-from-equilibrium conditions may contribute to many research fields, including active matter, adaptive dynamic systems and supramolecular and systems chemistry. We also believe that the results of this and preceding studies will further stimulate thinking on the fundamental mechanisms of emergent phenomena far from equilibrium. Moreover, our methodology has significant implications for nanotechnology since control of a large number of entities undergoing complex dynamics is generally thought to require a comparable number of control agents. This is not practical at very small scales, both due to limitations of available tools and also because random fluctuations become a dominant force, making traditional control all but impossible. One wonders whether our results can be exploited in this manner.

## Methods

**Experimental setup.** Colloidal solution of polystyrene nanoparticles (500 nm in diameter) was purchased from Microparticles GmbH. The solution is sandwiched in between two $\sim 150 \, \mu$m thick glass slides (Isolab Laborgeräte GmbH), where the edges of the cell are isolated. The specimen is placed on a Nikon inverted microscope. A custom-developed ytterbium-doped, all-normal dispersion fibre oscillator laser (Supplementary Fig. 1) is used to drive the self-assembly process that has a central wavelength of 1,030 nm and a spot size of $\sim 10 \, \mu$m in diameter ($1/e^2$). The operational principles of the laser system can be found in the references[48,49]. Repetition rate of the laser is reduced to 1 MHz by polarization-maintaining acousto-optic modulator. The laser pulse is dechirped down to 150 fs after the amplification. After dechirping, the pulse energy is reduced to 10 $\mu$J. To adjust the laser power during the experiments, the laser light is coupled to the transmission microscope system through a free-space acousto-optic modulator. A blue light source is used to illuminate the specimen to increase the resolution. Both the laser and the illumination light are coupled to the same optical path via a dichroic mirror and are focused onto the specimen with a 10 × , 0.25 NA high-power objective. The sample movement is controlled by a motorized, 2D translational stage (Thorlabs MLS203) with a minimum step size of $\sim 100$ nm. A 100 × , 1.3 NA oil immersion objective (Nikon CFI Plan Fluor ADH 100 × Oil) is used for imaging. A short-pass filter blocks laser light passing through the sample and the remaining light has been directed to a fast CMOS camera (optiMOS sCMOS).

**Control over bubble size and shape and density.** Laser pulses get nonlinearly absorbed in water and glass through multiphoton absorption[30] that depends on a power, $k$ (commonly, 2), of the laser intensity $I^k$. Laser fluence required to create an air bubble through multiphoton absorption is calculated to be 0.14 J cm$^{-2}$ matching well to the experimentally measured value of 0.13 J cm$^{-2}$. To create air bubbles in a controllable fashion, we have written a Matlab code, which we called the 'bubbleator', to instantly control the laser power. The experiments start when we instantly deposit high-energy laser pulses for a quick formation of an air bubble through boiling the water at that point. Then, the power is abruptly decreased by the bubbleator to prevent fast growth of the bubble. By further adjusting the laser power, we can enlarge or shrink the bubble and by spatially moving the laser beam, we can guide the 'hot steam' trapped inside the bubble to controllably change its shape. Using the bubbleator, we can create additional bubbles at desired locations. Controlling the number of bubbles and their sizes and shapes, we are able to define the boundary conditions and local nanoparticle density almost arbitrarily. This advanced control is crucial for observing multistable bifurcations during the experiments. A demonstration of the process can be viewed through Supplementary Movie 10.

**Detailed description of the numerical simulations.** Numerical simulations are performed via a commercial finite element code (COMSOL Multiphysics) for a computational area of 1 cm by 1 cm. The laser-induced Marangoni flow is simulated in 2D by solving Navier–Stokes equation self-consistently coupled to the convective heat equation (with a temperature-dependent body term) for an incompressible Newtonian fluid (laminar flow) as follows:

$$-\nabla.\eta\left(\nabla u+(\nabla u)^T\right)+\frac{\eta}{k}u+\nabla p=g\beta(T-T_c), \tag{1}$$

$$\nabla.u=0, \tag{2}$$

$$\nabla.\left(-k\nabla T+c_p\rho Tu\right)=0, \tag{3}$$

where $u$, $T$, $T_c$, $k$, $\eta$, $\rho$, $g$, $\beta$ and $p$ are convection velocity, temperature, reference temperature, heat conduction coefficient, viscosity constant, density of fluid, gravitational constant, thermal expansion coefficient and pressure, respectively. The first two equations describe the momentum and mass balance at the steady state. The third equation describes the heat balance, where the Boussinesq buoyant lifting term $T$ of ($c_p\rho Tu$) (the gravitational force due to density differences in the local environment) is used to couple the flow and the heat.

Navier–Stokes and convective heat equations are solved with a temperature-dependent body term. To represent a symmetrical flow, we assume that our 2D region is a deformation of 2D strip to annular patch with the centre at the point of laser heating introduced. Therefore, the buoyancy term is of the form,

$$\begin{pmatrix}F_x\\F_y\end{pmatrix}=\begin{pmatrix}g\rho\alpha(T-T_c)(x/r)\\g\rho\alpha(T-T_c)(y/r)\end{pmatrix}, \tag{4}$$

where $g$, $\rho$ and $\alpha$ are the gravitational constant, density of fluid and thermal expansion coefficient, respectively. Unit vectors in $x$ and $y$ direction are used to add a symmetric force with respect to origin where heat source introduced.

A bubble with a diameter of 50 μm is placed in the middle of the 1 cm by 1 cm cell. A boundary heat source is placed at the lower right quarter of the bubble to represent the heat energy delivered by the laser beam. No-slip boundary conditions are used for the rest. The effect of colloidal particles (polystyrene spheres with 500 nm diameters) to the process is calculated using Molecular Dynamics simulations considering the influence of the various forces acting on the particles, namely, drag, Brownian, thermophoretic, lift, gravitational forces as well as Lennard-Jones pair forces and added independently after the formation of microfluidic flow. Equation of motion for nanoparticle dynamics is integrated using Verlet algorithm[50,51].

**Detailed description of the toy model.** We first focus on the fluid dynamics in which the colloidal particles are dispersed. Under the influence of thermal energy deposited by the laser beam, the quasi-2D water layer exhibits Marangoni flow. The flow patterns are analysed numerically and discussed in the numerical simulations section above. The purpose of the toy model is to help understand formation of a self-assembled aggregate. Therefore, we focus our attention on the region where aggregation is expected to form and we make a number of simplifying approximations. The aggregation of colloidal particles can occur only in regions where the fluid flow speed is already low. As the particles aggregate, they influence the flow. In the low Reynolds number regime, a description of flow can be given by the Brinkman–Forchheimer equation,

$$\tau\frac{\partial}{\partial t}\vec{v}-\nu\nabla^2\vec{v}+\vec{v}=-\frac{\kappa A}{\mu}\vec{\nabla}P, \tag{5}$$

where $\tau$ is a time constant, $\mu$ is viscosity, $A$ is the cross-sectional area of the region and $P$ is pressure. Here, $\kappa$ is the permeability that depends on the aggregation, thus providing the coupling between the colloidal particles and fluid dynamics.

To describe the dynamics of the colloidal particles within the fluid, including the effect of Brownian motion, we write the Langevin equation of the following form:

$$m\ddot{r}(t)=-\frac{d}{dr}U(r)-\gamma\dot{r}(t)+\xi(t). \tag{6}$$

As it is commonly done, we drop the inertial term, which is a good approximation in the low Reynolds number regime, and we set $\gamma=1$,

$$\dot{r}(t)=-\frac{d}{dr}U(r)+\xi(t), \tag{7}$$

where $\xi(t)$ is white noise, describing the normalized Brownian force and $U(r)$ describes the potential energy arising from the mutual interactions of the particles. We are interested in what happens to the collection of colloidal particles, rather than an individual particle. This is succinctly described by the probability of density, $\rho(r,t)$, of finding a particle at a given location at a given time. This can be accomplished by switching to Fokker–Planck equation of the following general form:

$$\dot{\rho}(r,t)=\frac{\partial}{\partial t}\rho(r,t)=\frac{\partial}{\partial r}\left(\rho(r,t)\frac{\partial}{\partial r}U(r;r_1,r_2,\ldots,r_N)\right)+\frac{\partial^2}{\partial r^2}\rho(r,t). \tag{8}$$

Although we have written the equation above in 1D for clarity, its generalization to

higher dimensions is straightforward:

$$\dot{\rho}(\vec{r},t)=\vec{\nabla}\cdot\left[\vec{\nabla}(U(\vec{r};\vec{r}_1,\vec{r}_2,\ldots,\vec{r}_N))\rho(\vec{r},t)\right]+\nabla^2\rho(\vec{r},t). \tag{9}$$

Here, the first term on the right describes the drift and the second term describes diffusion due to Brownian motion. The exposition up to now is the same as that of a collection of Brownian particles subject to an external potential that are normally considered to be non-interacting. The situation of interest to us is their aggregation and self-assembly dynamics that must take into account nonlinear terms and, crucially, many-body effects. More specifically, $U(r;r_1,r_2,\ldots,r_N)$ depends on drift due to drag force of the fluid that, in turn, depends on the configuration of the particles, as well as (hard-sphere) interaction potential describing the bumping of a particle into another. Therefore, it depends, in principle, on all other particles, since they collectively influence the fluid flow, but at a minimum, strongly on their close-by neighbours. A detailed analysis is extremely complicated and beyond the scope of this study. For this reason, we resort to a semi-phenomenological approach. We begin by separating the two main contributors,

$$U(\vec{r};\vec{r}_1,\vec{r}_2,\ldots,\vec{r}_N)=U_{\mathrm{int}}(\vec{r};\vec{r}_1,\vec{r}_2,\ldots,\vec{r}_N)+U_{\mathrm{drag}}(\vec{r};\vec{r}_1,\vec{r}_2,\ldots,\vec{r}_N). \tag{10}$$

Next, we simplify the drag term $U_{\mathrm{drag}}(\vec{r};\vec{r}_1,\vec{r}_2,\ldots,\vec{r}_N)$ as $U_{\mathrm{drag}}\left(\vec{r};\phi_{\mathrm{aggregate}}\right)$ by assuming that it depends on the average aggregate size and density, but that the influence of individual colloidal particles on the fluid flow and therefore their influence on the drag force is negligible, which is a good approximation given the small size and large number of colloidal particles involved.

We now rewrite and simplify the coupled equations for the colloidal particle density and fluid flow:

$$\tau\frac{\partial}{\partial t}\vec{v}=\nu\nabla^2\vec{v}-\vec{v}-\frac{\kappa(\rho)A}{\mu}\vec{\nabla}P, \tag{11}$$

$$\frac{\partial}{\partial t}\rho=\nabla^2\rho+\vec{\nabla}\cdot\left[\vec{\nabla}(U_{\mathrm{int}})\rho-\epsilon\vec{v}\rho\right], \tag{12}$$

where we assumed the drag force to be simply proportional to the flow speed up to a proportionality constant, $\epsilon$. The form of these equations is formally the same as reaction–diffusion equations[36,52]. These coupled equations harbour the potential for aggregation zone to form as a result of the competing dynamics of aggregation due to flow and Brownian motion that is always dispersive.

These equations are still quite general and complex for a detailed examination. Thus, we now focus only on the aggregation zone. We introduce an order parameter, $\phi$, that refers to the filling ratio (fractional of area occupied by the colloidal particles for the aggregation zone) to quantify the level of aggregation inside this zone.

$$\phi=\iint\rho dA, \tag{13}$$

$\phi=0$ corresponds to an absence of particles and $\phi=1$ to the maximum packing allowed by geometrical constraints. The actual volumetric ratio that corresponds to maximum packing varies in the range of 0.395–0.476, depending on the lattice structure and assuming hard spheres, but such a distinction is not considered in this model.

The net fluid flux through this zone is similarly described by a single parameter, $\theta$, that is the scalar flow rate along the dominant direction of flow,

$$\theta=\oint_{\mathcal{S}}\vec{v}dA, \tag{14}$$

where $\mathcal{S}$ is the boundary of the aggregation zone (assuming the flow to be essentially 2D).

The fluid flux is caused by convective forces created by the nonlinear absorption of lasers pulses that typically occurs at a point outside of the aggregation zone. Thus, the flux thorough the semiporous region that the aggregate forms, depends on viscosity, pressure differences (due to the convective force) and permeability as follows:

$$\tau\dot{\theta}+\theta=-\frac{\kappa(\phi)A}{\mu}\frac{\Delta P}{L}, \tag{15}$$

where $\tau$ is a time constant, $\mu$ is viscosity, $A$ is the cross-sectional area of the zone, $L$ is the length of the zone along the direction of flux, $\Delta P$ is the pressure difference and $\kappa$ is the permeability. This result can be obtained from Navier–Stokes equations assuming laminar flow[53]. The dependence on $\phi$ arises from permeability. A commonly used expression that relates permeability to porosity is $\kappa=\frac{\Phi^3}{2S^2}$, where $\Phi$ is the porosity and $S$ is the specific surface[54]. Porosity is inversely proportional to the filling ratio, $\phi$. Therefore, if we introduce $F$ as the normalized convective force, the time evolution of flux can be compactly expressed as follows:

$$\tau\dot{\theta}+\theta=\frac{F}{\phi^3}, \tag{16}$$

The evolution of $\phi$ is much more complicated and certain simplifications are in order.

The influence of the diffusion term will always be towards reducing the aggregation (except when it is adjacent to a stronger point of aggregation, which we

do not consider). Given any situation of having higher particle density within the aggregation zone compared with the region outside (which we refer to as the background value), the influence of the Brownian motion will reduce it to the background value. This can be shown considering the aggregation as a perturbation of the form $\rho(\vec{r}, 0) = \rho_0 e^{-r^2/2\sigma}$. Then, considering only the influence of the diffusion term, that is, Brownian motion, we obtain

$$\phi(t) = \iint \rho \, dA = -2\pi \sqrt{\sigma} e^{-\frac{r^2}{2(\sigma+2t)}} \sqrt{\sigma + 2t}. \tag{17}$$

Therefore, we obtain for the rate of change of $\phi$ immediately after this perturbation and for an aggregation of zone size $L$,

$$\dot{\phi} = -\frac{2\pi}{\sigma} e^{-\frac{L^2}{2\sigma}} (\sigma + L^2) + O[t], \tag{18}$$

that is always a negative constant,

$$\langle \xi(t) \rangle_{\mathrm{rms}} \equiv \frac{2\pi}{\sigma} e^{-\frac{L^2}{2\sigma}} (\sigma + L^2). \tag{19}$$

The interaction term, in contrast, promotes aggregation, as it can trap new particles like a net. Following a similar Taylor expansion, we retain the first-order terms to obtain

$$\dot{\phi} = (\phi - \theta - \langle \xi(t) \rangle_{\mathrm{rms}}). \tag{20}$$

We expect this expression to describe the dynamics of the aggregation reasonably well when it is not yet dense. However, we need to take into account the fact that the aggregate growth must saturate as it cannot grow beyond the geometrically allowed maximum packing. Even the geometric limit cannot be attained in the presence of Brownian motion, since the jittery motion that it induces does not allow the colloidal particles to be permanently stationary and in contact with each other, thus not reaching the geometric limit. We incorporate these effects phenomenologically by adding a multiplicative term to the equation,

$$\dot{\phi} = (\phi - \theta - \langle \xi(t) \rangle_{\mathrm{rms}})(1 - \phi - \eta \langle \xi(t) \rangle_{\mathrm{rms}}). \tag{21}$$

Here, $\langle \xi(t) \rangle_{\mathrm{rms}}$ denotes the root-mean-square-averaged Brownian force acting on the colloidal particles within the aggregation zone and $\eta$ is a scaling parameter less than but close to 1, and is used merely to adjust the relative influence of Brownian motion in the two places it appears in this equation.

The first set of terms on the right, $(\phi - \theta - \langle \xi(t) \rangle_{\mathrm{rms}})$, describes the factors promoting and opposing aggregation. The first set of these terms describe the tendency of the aggregate to grow in proportion to its filling ratio, because the more particles there already are, the more likely for new particles to be scooped by the aggregate. However, fluid flux tends to drag particles away along its direction of flow, thereby opposing aggregation. The third term, Brownian motion, is directionless to first order and acts to disperse particles out of the zone, as discussed above. The second set of terms, $(1 - \phi - \eta \langle \xi(t) \rangle_{\mathrm{rms}})$, becomes significant only after a fairly dense aggregate forms and describes jamming. The particles are hard spheres and cannot occupy the same volume. Therefore, as the zone is increasingly full with particles, they tend to push each other away. The first set of terms is responsible for the initial rapid growth and the second set is responsible for the eventual saturation of the growth.

The coupled equations described above and reproduced below describe the evolution that can lead to formation of an aggregate or dispersal of all particles, depending on the starting conditions. For simplicity, $\eta$ is taken as 1 in what follows.

$$\dot{\phi} = (\phi - \theta - \langle \xi(t) \rangle_{\mathrm{rms}})(1 - \phi - \langle \xi(t) \rangle_{\mathrm{rms}}), \tag{22}$$

$$\dot{\theta} = \frac{F}{\tau \phi^3} - \theta/\tau. \tag{23}$$

This system has two fixed points: one is at $\phi^* = 1 - \langle \xi(t) \rangle_{\mathrm{rms}}$ and $\theta^* = \frac{F}{(1 - \langle \xi(t) \rangle_{\mathrm{rms}})^3}$. This is a stable node and corresponds to formation of an aggregate. The second fixed point is at $\phi^* = \theta^* + \langle \xi(t) \rangle_{\mathrm{rms}}$ and $\theta^* = \frac{F}{(\theta^* + \langle \xi(t) \rangle_{\mathrm{rms}})^3}$. While the latter expression is a fourth-order polynomial and can be solved exactly, the solution is not particularly illuminating. However, for reasonable values of $F$ and $\langle \xi(t)_{\mathrm{rms}} \rangle$, there is a single positive real root, thus physically acceptable for $\theta^*$. This root corresponds to a saddle point. The typical structure of the phase plane for this dynamic system is shown in Supplementary Fig. 4.

Here, the role of $\tau$ is limited to setting the relative rate at which fluid flow responds to changes in the configuration of the colloidal particles as described by $\phi$. As a simplifying approximation, we can assume $\tau$ to be extremely small and the reconfiguration to be instantaneous. In this limit, we obtain the well-known Darcy's equation for $\theta$ that becomes, $\theta = \frac{F}{\phi^3}$. Inserting this into the equation for $\phi$, the system is reduced to 1D,

$$\dot{\phi} = \left( \phi - \frac{F}{\phi^3} - \langle \xi(t) \rangle_{\mathrm{rms}} \right)(1 - \phi - \langle \xi(t) \rangle_{\mathrm{rms}}). \tag{24}$$

In this limit, the nature of the solutions is easier to visualize. The fixed points are $\phi^* = 1 - \langle \xi(t) \rangle_{\mathrm{rms}}$ and $\phi^* = \frac{F}{(\phi^*)^3} + \langle \xi(t) \rangle_{\mathrm{rms}}$. The second one is another fourth-order polynomial, again with a single positive real root. A typical case is shown in Fig. 1f of this study.

The present model is, by design, quite simple and cannot be expected to make quantitatively accurate predictions. However, it predicts several salient features observed in experiments. First, it predicts the existence of a critical filling ratio for formation of an aggregate. Under the influence of the constantly dispersive Brownian motion and the convective force, which not only brings in particles, but also drags them away, a sustained aggregate cannot be formed unless a critical filling ratio corresponding to the saddle point in the 2D system and the unstable fixed point in the 1D system is exceeded. This explains why the colloidal aggregates do not form spontaneously, but require prearrangement: we experimentally achieve this through the creation of an air bubble that forms a physical barrier for fluid flow and effectively sets the initial value of $\phi$ to a point above this critical value, after which the aggregate freely grows, until it reaches the stable fixed point. In addition, the toy model shows that the stable fixed point is destroyed and no aggregation is possible if the convective force is too strong—convection simply sweeps away all the particles (as also predicted by the simulations, Supplementary Fig. 2f for high $\Delta T$ values). The same is true for Brownian motion.

**Data analysis and image processing.** Coding and image processing for data analyses are performed using Matlab. Circle Hough transform[55] is used to detect the particles after a pre-processing step. Four descriptive analyses are made after the particle detection step; namely, number fluctuations, Lindemann parameter, reciprocal lattice analysis and pair correlation function. Detailed information on detection algorithms and parameter extraction methods is provided in the Supplementary Method 3 document.

**Data availability.** The data sets generated during and/or analysed during the current study are available from the corresponding author on reasonable request.

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

## Acknowledgements

This work was supported partially by the European Research Council (ERC) Consolidator Grant ERC-617521 NLL and TÜBITAK under project 115F110.

## Author contributions

S.I. designed the research and interpreted the results with help from F.Ö.I, O.G. and O.T. Experiments were performed by S.I., with help from G.M., Ö.Y. and I.P. Analytical model was developed by F.Ö.I. Numerical simulations were performed by G.B.A., S.I. and O.G. Image processing analyses were conducted by G.M.

## Additional information

**Competing interests:** The authors declare no competing financial interests.

