## [Peer Review File · Nature Communications]

Reviewers' comments:

Reviewer #1 (Remarks to the Author):

The paper by Ilday et al. describes an elegant piece of work in which an interplay between the flow of liquid, Brownian motion, and liquid/air boundary results in a dynamic clustering of polystyrene particles. The hydrodynamic flow is maintained by a temperature gradient from laser pulses. Therefore, the system is out-of-equilibrium. The clusters of polystyrene particles are dynamic and fall apart when the energy source is removed. Although this system involves hydrodynamic flow, it is rather simple and intuitively understandable through feedback between flow, Brownian motion, and the size of the cluster. The toy model, which was elegantly derived, gives a good qualitative understanding of the dynamics of this system. Thus, the work is unique for the field in the sense that it demonstrates a reach dynamic behaviour of the microscale structures, involves flow, but can be understood through a simple model. I especially notice the competition between two lattices. It is a fascinating effect that demonstrates a bistability in the system (video 5). Because of its simplicity, the work will stimulate thinking in the field of active colloids, but also in the fields of dynamic materials, supramolecular and systems chemistry. The experimental set-up provides opportunities for future experiments where the size, the shape of the particles, and properties of the liquid will be varied.

There are, however, a few points that can improve the manuscript.

(i) The conclusions should tell clearly what do we specifically learn from the paper? In the introduction, the problem statement sounds "current understanding of the fundamental mechanisms and availability of experimental tools to test emerging theories are lacking".

Therefore, a reader expects to find new insights about fundamental mechanisms or explanation of the advantages of a new experimental tool in the conclusions.

(ii) Claims about self-regulation and self-healing, from my opinion, need more experimental support. Self-regulation means that system can adapt to a change in an environmental parameter. Video 4, as I can see, does not show any changes in environmental parameters and responses to these changes. Would be good, if authors can demonstrate a response of the system to a change in an environmental parameter. The similar situation is with self-healing. Video 6 demonstrates more the dynamic fluctuation in the structure than self-healing. I suggest an experiment where the structure is damaged by an external force.

(iii) About the title. Emergent behaviours are complex; probably no need for both words. Also, would be good to make the title more specific that it is now.

(iv) Some terminology issues. Authors use the term "life-like properties" in the abstract. This term is unclear for me. The most important property of life is an ability to undergo evolution. The system that is described in this work can't undergo evolution. On page 1, "to test emerging theories". Emerging theories of what?

(v) Description of some extra experimental details and some additional explanations will benefit the broad readership of Nature Communications. First, where is the gas for the bubbles coming from? Is it overheated water vapors? Why they do not collapse immediately after removal of heating? Is it air that was dissolved in water? Do flow simulations include a bubble? If yes, it should be shown on the video. What is the $U(r)$ in the model? It has units of energy, but readers will benefit from more additional explanations.

In summary, the manuscript will be interesting for the specialists and the broad readership. It should be suitable for the publication when the points mentioned above are addressed.

Sincerely yours,
Sergey N. Semenov
Harvard University

Reviewer #2 (Remarks to the Author):

CONTENTS: In this paper, a two-dimensional cavity for polystyrene beads is created by a stacking of two glass plates spaced by two micrometres, which is four particle diameters. An air bubble is created together with strong thermal gradients by a laser spot. The thermal gradients induce a gradient in the temperature of Brownian motion and of course a mesoscopic particle convection that forms a regular pattern around the laser spot. By this, crystal growth is induced at the 2-D bubble interface close to the laser spot, and square, hexagonal and stacked hexagonal bilayers (manifested as a pseudo-honeycomb) can be seen to grow to a certain scale at steady state. The maintained steady-state size of the half-disked crystals seems to be determined by the size of the 2-D bubble and thus the laser intensity. Upon putting the laser down, the crystal becomes a glassy aggregate and dissolves. It is clear that this is a nice example of dispersive crystal growth and dissolution maintained in a steady state far from equilibrium. On irregular line shapes, for instance at the inclination between two connected bubbles, a pseudo-honeycomb and moire lattice form a biphasic system, and it is shown that one can be converted in the other in a reversible way. Finally, it is also shown that the particle density fluctuations scale with N when the laser creates far from equilibrium conditions.

EVALUATION: MAJOR REVISION REQUIRED

This is a very nice paper on far from equilibrium 2-D crystal growth ("active self-assembly" according to G. Whitesides, *Science* 2002), and from this viewpoint I believe it is of interest for the readership of *Nature communications*. I have problems with the language used, especially with the notion "self-replication", which I appreciate as simple crystal growth and with the connection – solely in the abstract – "minimum requirements for life-like properties from non-living system". This should be much better supported in the text or deleted.

Minor technical comments:

1. Upon excitation with a laser spot, a depletion layer is seen around the air bubble, while on one specific place the crystal grows to a certain size and is maintained. The depletion layer is not mentioned nor explained. Please do.
2. The crystal initially growing at the line inclination of the two bubbles is not a hexagonal monolayer, but a hexagonal bilayer which is apparent as a "honeycomb". See C. B. Murray, *Nature* 2015. Genuine honeycombs can also be formed at an interface, see M. Boneschanscher et al., *Science* 2014.
3. The so-called self replication in video 4 and figure 4a. I only see the growth of a single aggregate, not the birth of a second aggregate from the existence of the first. Hence, is this true self-replication?
4. Figure 1: the surface density Φ is not defined in the caption.
5. The laser light can charge discharge the particles and in this way influence, the particle interactions and thus the crystallisation. The authors should check if this is the case with the polystyrene particles used here.

RESPONSES TO THE REVIEWER COMMENTS

Reviewer #1:

The paper by Ilday et al. describes an elegant piece of work in which an interplay between the flow of liquid, Brownian motion, and liquid/air boundary results in a dynamic clustering of polystyrene particles. The hydrodynamic flow is maintained by a temperature gradient from laser pulses. Therefore, the system is out-of-equilibrium. The clusters of polystyrene particles are dynamic and fall apart when the energy source is removed. Although this system involves hydrodynamic flow, it is rather simple and intuitively understandable through feedback between flow, Brownian motion, and the size of the cluster. The toy model, which was elegantly derived, gives a good qualitative understanding of the dynamics of this system. Thus, the work is unique for the field in the sense that it demonstrates a reach dynamic behavior of the microscale structures, involves flow, but can be understood through a simple model. I especially notice the competition between two lattices. It is a fascinating effect that demonstrates a bistability in the system (video 5). Because of its simplicity, the work will stimulate thinking in the field of active colloids, but also in the fields of dynamic materials, supramolecular and systems chemistry. The experimental set-up provides opportunities for future experiments where the size, the shape of the particles, and properties of the liquid will be varied.

We thank the Reviewer for this very positive review. We appreciate that the Reviewer has caught the core message of our study, which is extremely important and encouraging for us. We plan to follow the advice of the Reviewer and apply our methodology to different type of materials with various sizes and shapes to further uncover the self-assembly dynamics in far-from-equilibrium systems. In fact, our efforts are already under way.

There are, however, a few points that can improve the manuscript.

(i) The conclusions should tell clearly what do we specifically learn from the paper? In the introduction, the problem statement sounds “current understanding of the fundamental mechanisms and availability of experimental tools to test emerging theories are lacking”. Therefore, a reader expects to find new insights about fundamental mechanisms or explanation of the advantages of a new experimental tool in the conclusions.

We thank the Reviewer for pointing out this issue. The conclusion part is revised as follows:

“In conclusion, we provide the simplest experimental platform to date for far-from-equilibrium self-assembly to investigate and control a rich set of complex phenomena. We demonstrated collective control of large groups of nanoparticles using only two parameters, the laser power and the beam position, which act at much larger spatial ($\sim 10 \mu\text{m}$) and temporal (few seconds) scales than those of the individual particles ($0.5 \mu\text{m}$, milliseconds) being controlled. Although we worked with 500-nm particles allowing for real-time optical imaging, the scaling of the thermal gradients and strength of Brownian motion with beam and particle size, respectively, suggest the possibility of controlled self-assembly of sub-10 nm nanoparticles using a diffraction-limited laser beam ($\sim 250 \text{ nm}$). In principle, the methodology can also be applied to other types of materials, nonliving and living alike, with

different shapes and properties. The possibility of studying such materials under far-from-equilibrium conditions can contribute to many research fields, including active matter, adaptive dynamic systems, and supramolecular and systems chemistry. We also believe that the results of this and preceding studies will further stimulate thinking on the fundamental mechanisms of emergent phenomena far from equilibrium. Moreover, our methodology has significant implications for nanotechnology since control of a large number of entities undergoing complex dynamics is generally thought to require a comparable number of control agents. This is not practical at very small scales, both due to limitations of available tools and also because random fluctuations become a dominant force, making traditional control all but impossible. One wonders whether our results can be exploited in this manner.”

(ii) Claims about self-regulation and self-healing, from my opinion, need more experimental support. Self-regulation means that system can adapt to a change in an environmental parameter. Video 4, as I can see, does not show any changes in environmental parameters and responses to these changes. Would be good, if authors can demonstrate a response of the system to a change in an environmental parameter.

Done. In response to the Reviewer's request, we have performed an additional experiment to demonstrate self-regulation of the aggregate in a highly dynamic environment (Video 4b): The video starts with an already formed aggregate at the boundary of a small bubble ($t = 0$ s). By increasing the laser power, we initiate the growth of the bubble and the aggregate size ($t = 15$ s). Then, by moving the laser beam, we enlarged the bubble but the average size of the aggregate is maintained during this period ($t = 82$ s). Even if we further accelerate the fluid flow, self-regulation mechanism is active and prevents further growth of the aggregate ($t = 142$ s). Please note that the solution is highly dense and at such solutions one would expect jamming of the large numbers of incoming particles at the aggregate boundary and cause further growth of the aggregate size. However, strong Brownian motion of the particles (negative feedback) balance out this tendency and the aggregate self-regulates and maintains its size. Examples to such self-regulation mechanisms induced by feedback loops can be found in references¹⁻³. We have deliberately changed the focus of the objective in order to make sure that the aggregate size does not change from one layer to another (105 s $< t < 130$ s). Finally, by repositioning the laser beam and decreasing the laser power, we start to shrink the bubble size and show that self-regulation still holds ($t = 143$ s).

Apart from this experiment, we believe that our claims for Video 4 (now Video 4a) are, in fact, correct, but we did not explain them clearly enough and also because the dynamic changes were a little subtle. We hope that the explanations given below will clarify.

Left frame (diluted colloidal solution): In this video, an already formed aggregate at the upper wedge of the two touching bubbles is actively regulating its average size in a dynamic environment. The variable parameters in this dynamic environment are the fluid flow and the number of particles that are constantly being carried to the vicinity of the aggregate by the flow. The flow pattern is affected by the flux at the bottom wedge of the two touching bubbles, which can be recognized from the changes in

the direction of motion of the particles. The incoming particles that are being carried towards the aggregate are expected to join in and further enlarge the aggregate, however, this does not happen since strong Brownian motion of the particles (negative feedback) regulates this tendency and the overall aggregate size is maintained.

Right frame (dense colloidal solution): We specifically used a highly dense solution to prove that the self-regulation is a result of the balancing act of the negative feedback. In such a dense solution, one expects jamming of the particles and further growth of the aggregate. However, strong Brownian motion regulates this effect and helps maintain the overall aggregate size. In this video, flow pattern is also changing by promoting the bubble growth via increasing the laser power. Upon growing ($t = 9$ s), the bubble pushes the particles away from its upper boundary towards the aggregate. This clearly changed the flow pattern evidenced from the change in the particles direction of motion and their speed.

Accordingly, we have revised the manuscript, the details of which are given below:

“Moreover, these aggregates can self-regulate in a dynamic environment as shown in Videos 4a and 4b: Video 4a shows that the aggregates in a diluted (left frame) and in a dense (right frame) colloidal solution are self-regulating to maintain their overall size in a dynamical environment. Left frame shows that the flow constantly carries new particles towards the aggregate. These particles are expected to join in and further enlarge the aggregate, yet, this does not happen since strong Brownian motion of the particles (negative feedback) regulates this tendency and the overall aggregate size is maintained. Similarly, the right frame shows no increase in aggregate size even in a highly dense solution, where jamming of the particles are expected to cause further growth of the aggregate. However, negative feedback again regulates this effect and helps maintain the overall aggregate size. Video 4b shows self-regulation in a more visibly dynamic environment: The video starts with an already formed aggregate at the boundary of a small bubble ($t = 0$ s). By increasing the laser power, we initiate the growth of the bubble and the aggregate size ($t = 15$ s). Then, by moving the laser beam, we enlarge the bubble but the average size of the aggregate is maintained during this period ($t = 82$ s). Even if we further accelerate the fluid flow, self-regulation mechanism is active and prevents further growth of the aggregate ($t = 142$ s). We then deliberately change the focus of the objective in order to verify that the aggregate size does not change from one layer to another (105 s $< t < 130$ s). Finally, by repositioning the laser beam and decreasing the laser power, we shrink the bubble and show that self-regulation still holds ($t = 143$ s).”

The similar situation is with self-healing. Video 6 demonstrates more the dynamic fluctuation in the structure than self-healing. I suggest an experiment where the structure is damaged by an external force.

Done. In response to the Reviewer's request, we have performed an additional experiment to demonstrate self-healing of the aggregate (Video 6b). The video starts with a large square lattice. In order to introduce lattice imperfections, we start to move the laser beam inside the bubble. This

distorts the shape of the bubble and enlarges it ($t = 33$ s). Then, we move the laser beam again and introduce a second bubble to squeeze the pattern in a wedge ($t = 34$ s), which totally disrupts the square lattice ($t = 49$ s). Finally, we set free the squeezed pattern by detaching the second bubble from the first one (by moving the laser beam) and the square pattern self-heals the stress induced defects and fully recovers.

Similar to the self-regulation case, we stand by our claims for Video 6 (now Video 6a). As the Reviewer pointed out, Video 6a demonstrates dynamic fluctuations. These dynamic fluctuations insert a number of lattice imperfections (mainly point and line defects) to the crystal structure. It is elimination (annealing out) of these imperfections to maintain the overall hexagonal pattern that constitutes self-healing which are constantly being annealed out of the structure to maintain the overall hexagonal pattern. We understand this was not clear in the manuscript. We thank the Reviewer for pointing out this potentially confusing issue and we have added necessary remarks in the text accordingly, the details of which are given below:

“Similarly, Video 6a and Fig. 4c show self-healing under weak perturbations, where the hexagonal lattice repeatedly anneals out the lattice imperfections (orange dotted line shows a line defect and green circle shows a point defect) to maintain its original pattern. In Video 6b, we also show self-healing ability in a square lattice that has been heavily damaged: The video starts with a large square lattice. In order to introduce lattice imperfections, we start to move the laser beam inside the bubble. This results in distorting the shape of the bubble and enlarging it ($t = 33$ s). Then, we move the laser beam again and introduce a second bubble to squeeze the pattern in a wedge ($t = 34$ s), which totally disrupts the square lattice ($t = 49$ s). Finally, we release the squeezed pattern by detaching the second bubble from the first one (by moving the laser beam) and the square pattern self-healed the stress induced defects and fully recovered.”

(iii) About the title. Emergent behaviors are complex; probably no need for both words. Also, would be good to make the title more specific that it is now.

Agreed. The new title is: *“Rich complex behavior of self-assembled nanoparticles far from equilibrium”*.

(iv) Some terminology issues. Authors use the term “life-like properties” in the abstract. This term is unclear for me. The most important property of life is an ability to undergo evolution. The system that is described in this work can’t undergo evolution.

Done. We replaced the “life-like properties” term with “complex behavior” in the abstract. The corresponding sentence now reads as follows: *“... What are the minimum requirements for emergence of complex behavior from non-living systems? ...”*

On page 1, “to test emerging theories”. Emerging theories of what?

We have corrected the sentence as follows: “*However, current understanding of the fundamental mechanisms and availability of experimental tools to test emerging theories on the subject are lacking.*”

(v) Description of some extra experimental details and some additional explanations will benefit the broad readership of Nature Communications.

Done. In response to the Reviewer’s request, we have moved the following sections from Supplementary Information document to the Methods Section of the manuscript:

- Details on the bubble formation and its control mechanisms are described in the Supplementary Information document (Section 2),
- Details of numerical analyses are given in the Supplementary Information document (Section 3),
- Details of analytical model is given in the Supplementary Information document (Section 4).

First, where is the gas for the bubbles coming from? Is it overheated water vapors? Why they do not collapse immediately after removal of heating? Is it air that was dissolved in water?

Yes, it is the air dissolved in the water. The experiments start when we instantly deposit high-energy laser pulses for a quick formation of an air bubble through boiling the water at the solid (glass)-liquid (colloidal solution) interface. This is due to the fact that the laser pulses get nonlinearly absorbed in water and glass through multi-photon absorption. Then, the power is abruptly decreased to prevent fast growth of the bubble. By further adjusting the laser power, we can enlarge or shrink the bubble and by spatially moving the laser beam, we can guide the “hot steam” trapped inside the bubble to controllably change its shape (Described in the Methods Section now).

Once the air bubble is formed, it is sustained by the balance between the evaporation of water at the bubble surface and the condensation of the vapor at the liquid-vapor interface inside the bubble⁴⁻⁶. The bubble can remain stable for hours this way since we also have convection heat transfer due to the Marangoni flow in our system.

Do flow simulations include a bubble? If yes, it should be shown on the video.

Yes. The bubble is described in the captions of Fig. 1b and Video 1. It is also described in the Supplementary Information Section 3, page 4. The bubble is shown at the center in Video 1.

What is the $U(r)$ in the model? It has units of energy, but readers will benefit from more additional explanations.

$U(r)$ describes the potential energy arising from the mutual interactions of the particles. We have modified the relevant section to clarify.

In summary, the manuscript will be interesting for the specialists and the broad readership. It should be suitable for the publication when the points mentioned above are addressed.

We sincerely thank the Reviewer for many insightful comments. All changes we have made to the original text can be seen in the marked-up version of the revised manuscript. We hope that the Reviewer will find the revised manuscript to be much clearer.

References (for Reviewer 1):

1. Heuser, T., Steppert A.-K., Lopez, C. M., Zhu, B., Walther, A. Nano Lett. 15, 2213-2219 (2015).
2. Der, R. & Martius, G. The playful machine: Theoretical foundation and practical realization of self-organizing robots. Springer-Verlag Berlin Heidelberg (2011).
3. Baskaran, A. & Marchetti, M. C. Eur. Phys. J. E. 35, 95 (2012).
4. Zou, A., Chanana, A., Agrawal, A., Wayner Jr., P. C., Maroo, S. C. Sci. Rep. 6, 20240 (2016).
5. Liu, X., Guo, D., Xie, G., Liu, S. & Luo, J. Appl. Phys. Lett. 101, 211602 (2012).
6. Petrovic, S., Robinson, T., Judd, R. L. Int. J. Heat Mass Tran. 47, 5115–5128 (2004).

Reviewer 2:

CONTENTS: In this paper, a two-dimensional cavity for polystyrene beads is created by a stacking of two glass plates spaced by two micrometres, which is four particle diameters. An air bubble is created together with strong thermal gradients by a laser spot. The thermal gradients induce a gradient in the temperature of Brownian motion and of course a mesoscopic particle convection that forms a regular pattern around the laser spot. By this, crystal growth is induced at the 2-D bubble interface close to the laser spot, and square, hexagonal and stacked hexagonal bilayers (manifested as a pseudo-honeycomb) can be seen to grow to a certain scale at steady state. The maintained steady-state size of the half-disked crystals seems to be determined by the size of the 2-D bubble and thus the laser intensity. Upon putting the laser down, the crystal becomes a glassy aggregate and dissolves. It is clear that this is a nice example of dispersive crystal growth and dissolution maintained in a steady state far from equilibrium. On irregular line shapes, for instance at the inclination between two connected bubbles, a pseudo-honeycomb and moire lattice form a biphasic system, and it is shown that one can be converted in the other in a reversible way. Finally, it is also shown that the particle density fluctuations scale with N when the laser creates far from equilibrium conditions.

EVALUATION: MAJOR REVISION REQUIRED

This is a very nice paper on far from equilibrium 2-D crystal growth (“active self-assembly” according to G. Whitesides, Science 2002), and from this viewpoint I believe it is of interest for the readership of Nature communications.

We thank the Reviewer for this very positive review.

I have problems with the language used, especially with the notion “self-replication”, which I appreciate as simple crystal growth and with the connection – solely in the abstract – “minimum requirements for life-like properties from non-living system”. This should be much better supported in the text or deleted.

We regret to hear that the Reviewer did not find our claims on “self-replication” convincing, which shows that we have not explained them well enough. Our reasoning was as follows:

In our study, self-replication is defined as the act of a structure making identical copies of itself on an adjacent region as introduced for cellular automata by von Neumann¹ and used in many other contemporary works as well²⁻⁵. If the system was in thermodynamic equilibrium or near-equilibrium, we would not have referred to it as self-replication, but state that the crystal merely grows. In our case, growth of the aggregate is not the result of relatively simple dynamics, but rather one of qualitatively distinct outcomes, which include conversion to multiple other patterns: In a dynamic system with many kinetic traps, different patterns can co-exist and compete, where propagation of the replication information for one of the species must lead to the degradation of the rest of the competing species (by destabilizing their kinetic traps) and amplification of the remaining species (by

promoting one stable kinetic trap)²⁻⁹. Such selection–amplification cycles are considered to be very important for self-replicating complex materials. In addition, we note that self-replication is almost always accompanied by an autocatalysis process¹⁻⁹, which we also show.

Accordingly, we have revised the manuscript, and the relevant section is reproduced below:

“In other words, the hexagonal lattice dies and the Moiré pattern survives the competition and self-replicates. Here, self-replication refers to a structure making identical copies of itself on an adjacent region⁶⁻⁸ as described for cellular automata by von Neumann³. If the system was in thermodynamic equilibrium or near-equilibrium, we would not have regarded to it as self-replication, but as crystal growth. In our case, growth of the aggregate is one of many possible, qualitatively distinct outcomes, which include conversion to multiple other patterns: In a dynamic system with many kinetic traps, different patterns can co-exist and compete, where propagation of the replication information for one of the species must lead to the degradation of the rest of the competing species (by destabilizing their kinetic traps) and amplification of the remaining species (by promoting one stable kinetic trap)^{8,40-45}. Moreover, we also show self-replication of a “daughter” aggregate from the “mother” aggregate in Video 5b: Left frame shows that a bubble forms and Marangoni flow drags the particles towards its boundary to form an aggregate. Then, a second bubble forms and separates the aggregate into two aggregates with the same pattern. Similarly, the right frame shows that part of an already formed large aggregate is being detached and carried to the boundary of another bubble to form the same pattern. Furthermore, our observation is not limited to the hexagonal lattice and the Moiré pattern; we observed self-replication of square lattice and its competition with the hexagonal lattice (Fig. 4b).”

Minor technical comments:

1. Upon excitation with a laser spot, a depletion layer is seen around the air bubble, while on one specific place the crystal grows to a certain size and is maintained. The depletion layer is not mentioned nor explained. Please do.

Done. Accordingly, we have revised the manuscript, the details of which are given below:

“Due to this drag force, a region that is fully depleted of particles forms around the bubble.”

We thank the Reviewer for noticing and pointing out this issue.

2. The crystal initially growing at the line inclination of the two bubbles is not a hexagonal monolayer, but a hexagonal bilayer which is apparent as a “honeycomb”. See C. B. Murray, Nature 2015. Genuine honeycombs can also be formed at an interface, see M. Boneschanscher et al., Science 2014.

We agree with the Reviewer that the initial structure in Fig. 4a is a hexagonal bilayer. However, we respectfully disagree that the structure is a “honeycomb” arrangement of the bilayer. Because, the top and bottom layers of the hexagonal lattice are located on top of each other without any spatial

shifting (as in AA stacking). Therefore, the lattice cannot be in a honeycomb arrangement but should be hexagonal.

In order to prove that, we have performed additional image processing analysis (Fig. R1): The images have been captured from Video 5 (now Video 5a), where the focus of the objective has been intentionally changed ($11 \text{ s} < t < 18 \text{ s}$) to show that we have a bilayer lattice. As can be seen from the video and from Fig. R1, upon changing the focus of the objective, the bright circles (Fig. R1a) turn into dark circles (Fig. R1b) and they preserve their spatial positions. This means that the bright and dark circles in Layer 1 and 2 are polystyrene particles and they are located on top of each other (as in AA stacking): (i) the red circle in Layer 2 shows a defect, whereas Layer 1 has no defects, which clearly distinguishes one layer from the other (ii) by overlapping the two layers (white + white = white, black + white = white, white + black = white, and black + black = black) as shown in Fig. R1c, we show that there are no dark circles indicating that the layers are in AA stacking, (iii) close-up images taken from the same field of view of both layers proves that the particles are located on top of each other (Fig. R1d).

Figure R1. Images captured from Video 5a shows polystyrene particles **(a)** as bright circles on the top layer and **(b)** as dark circles on the bottom layer. **(c)** Overlapping image of the top and bottom

layers show that the particles on both layers are located exactly on top of each other. **(d)** Images taken from the same field of view show that the particles in both layers are located on top of each other.

We have added this additional analyses to the Supplementary Information document along with the two references that the Reviewer pointed out.

3. The so-called self replication in video 4 and figure 4a. I only see the growth of a single aggregate, not the birth of a second aggregate from the existence of the first. Hence, is this true self-replication?

In response, we have performed two additional experiments to demonstrate self-replication of a “daughter” aggregate from the “mother” aggregate as shown in Video 5b:

Left frame: In this video, a bubble forms and Marangoni flow drags the particles towards the bubble boundary to form an aggregate. Then, a second bubble forms and separate the aggregate into two aggregates with same pattern.

Right frame: In this video, part of an already formed large aggregate is being detached and carried to the boundary of another bubble to form the same pattern.

We hope that the Reviewer will find these new results convincing. We thank the Reviewer for providing us the opportunity to better explain our results.

Accordingly, we have revised the relevant paragraph of the manuscript, which was given above (page 8 of this document).

4. Figure 1: the surface density PHI is not defined in the caption.

Done. PHI is defined in the caption in the revised manuscript.

5. The laser light can charge discharge the particles and in this way influence, the particle interactions and thus the crystallisation. The authors should check if this is the case with the polystyrene particles used here.

We are sure that this is not the case in our study: First, the laser beam is not directly interacting with the particles but it is confined inside the bubble so particles can not be charged by the laser. Second, we do not have any optical trapping/tweezing effect in our system since we use femtosecond laser pulses at an average power too low for such effects, even if the particles were positioned under the beam (this was verified).

We thank the Reviewer very much for providing us with an opportunity to better explain our ideas and methodology. All changes we have made to the original text can be seen in the marked-up version of the revised manuscript. We hope that the Reviewer will find the revised manuscript to be much clearer.

References (for Reviewer 2):

1. Von Neumann, J., Burks, A. W. Theory of Self-Reproducing Automata. Univ of Illinois Press, Urbana, IL (1966).
2. Zeravcica, Z., Brenner, M. P. PNAS 4, 111, 1748–1753 (2014).
3. Bissette, A. J. & Fletcher, S. P. Angew. Rev. 52, 12800-12826 (2013). Mattia, E. & Otto, S. Nature Nanotech. 10, 111-119 (2015).
4. Pugatch, R. PNAS 112, 2611-2616 (2015).
5. Bottero, I., Huck, J., Kosikova, T., Philp, T. JACS 138, 6723-6726 (2016).
6. England, J. L. J. of Chem. Phys. 139, 121923 (2013).
7. Saric, A., Buell, A. K., Meisl, G., Michaels, T. C. T., Dobson, C. M., Linse, S., Knowles, T. P. J., Frenkel, D. Nature Phys. 12, 874-882 (2016).
8. Mann, S. Angew. Chem. 47, 5306-5320 (2008).
9. Zhang, R., Walker, D. A., Grzybowski, B. A., de la Cruz, M. O. Angew. Chem. 53, 173-177 (2014).

REVIEWERS' COMMENTS:

Reviewer #1 (Remarks to the Author):

I believe that authors fully addressed my points. The new version of the manuscript is clearer than the original one and can be published as is.

Reviewer #2 (Remarks to the Author):

(This reviewer made confidential comments to the editor only.)